# Rest Induces a Distinct Transcriptional Program in the Nervous System of the Exercised *L. stagnalis*

**DOI:** 10.3390/ijms26146970

**Published:** 2025-07-20

**Authors:** Julian M. Rozenberg, Dmitri Boguslavsky, Ilya Chistopolsky, Igor Zakharov, Varvara Dyakonova

**Affiliations:** Koltzov Institute of Developmental Biology of the Russian Academy of Sciences, 119334 Moscow, Russia; boguslavsky@rambler.ru (D.B.); iszakharov@yandex.ru (I.Z.)

**Keywords:** exercise, rest, nervous system, *L. stagnalis*, neurodevelopmental diseases, autism

## Abstract

In the freshwater snail *L. stagnalis*, two hours of shallow water crawling exercise are accompanied by the formation of memory, metabolic, neuronal, and behavioral changes, such as faster orientation in a novel environment. Interestingly, rest following exercise enhances serotonin and dopamine metabolism linked to the formation of memory and adaptation to novel conditions. However, the underlying transcriptional responses are not characterized. In this paper, we show that, while two hours of forced crawling exercise in *L. stagnalis* produce significant changes in nervous system gene expression, the subsequent rest induces a completely distinct transcriptional program. Chromatin-modifying, vesicle transport, and cell cycle genes were induced, whereas neurodevelopmental, behavioral, synaptic, and hormone response genes were preferentially repressed immediately after two hours of exercise. These changes were normalized after two hours of the subsequent rest. In turn, rest induced the expression of genes functioning in neuron differentiation and synapse structure/activity, while mitotic, translational, and protein degradation genes were repressed. Our findings are likely relevant to the physiology of exercise, rest, and learning in other species. For example, chronic voluntary exercise training in mice affects the expression of many homologous genes in the hippocampus. Moreover, in humans, homologous genes are pivotal for normal development and complex neurological functions, and their mutations are associated with behavioral, learning, and neurodevelopmental abnormalities.

## 1. Introduction

Environmental changes can force animals to migrate, presuming more intense locomotion or physical exercise than under normal conditions. Exercise, in its turn, is known to activate the brain functions in both vertebrates and invertebrates [1,2,3,4,5,6,7,8,9,10,11]. Voluntary running improves memory, forces neurogenesis, facilitates decision-making, and decreases anxiety and depression, diminishing the adverse influence of stress [1]. These changes could play an important role in the adaptation of the nervous system to changing environment and life conditions [2]. In vertebrates, these benefits seem to be mediated by several brain neuromodulatory and neurotrophic systems [3]. More recently, changes in brain gene expression and open chromatin state were documented in rats and mice, which extend even to the next generations of trained animals [4,5]. Emerging data from research on various protostomes suggest that the beneficial effects of species-specific intense locomotion on the brain may have occurred at the early stages of animal evolution [2]. For example, in the nematode *C. elegance*, motor activity (swimming) facilitates learning and memory and has a protective effect against neurodegenerative diseases [6]. In insects, intense locomotion increased endurance and improved sleep and the feeding behavior of Drosophila [7], and improved orientation and enhanced aggressiveness and resistance to some disturbing stimuli in *G. bimaculatus* [2]. Many of these effects have previously been described in mammals as the beneficial results of running, suggesting certain similarity among distantly related species. These discoveries opened new perspectives for elucidating the molecular mechanisms underlying the influence of exercise on brain functions and adaptation to novelty, as many invertebrates are excellent experimental models.

In the freshwater snail *Lymnaea stagnalis*, intense muscular crawling in low water is accompanied by the formation of memory, facilitating subsequent behavior in a novel completely dry arena [8,9,10,11]. Snails, after a single bout of physical exercise, make faster decisions and have a higher speed of locomotion in a novel environment [8]. These behavioral changes correlate with serotonin neuron excitation after exercise, followed by a dopamine-dependent decrease in the rested-after-exercise animals [11]. The comparison of behavior, neuronal activity, and serotonin metabolism in the exercised, control, and rested-after-exercise animals suggests that rest after exercise is a remarkable state of the nervous system [10,11]. It is different from the state of control, non-exercised animals and showed an even greater difference in comparison to the animals assessed immediately after exercise [10]. Transcriptional changes co-occurring with exercise and rest are not known. We hypothesized that exercise and rest induce transcriptional activation and repression and performed gene expression profiling and annotation to figure out the possible functional impact of genes associated with exercise and rest after exercise in snails (Figure 1). We observed that exercise activated the expression of genes associated with chromatin remodeling and epigenetic regulation, while a number of genes related to neuronal communication and behavior were repressed. In contrast, rest activated the expression of developmental genes associated with neuron projection development and synapse formation and re-activates behavioral genes. We propose that the transcriptional switch observed after rest may play a significant role in physical and cognitive adaptations to novel conditions.

## 2. Results

### 2.1. Transcriptome Assembly and Identification of Proteins Homologous to the Coding Sequences of L. stagnalis

The complete sequence and annotation of the *L. stagnalis* transcriptome is not available yet. Therefore, to estimate gene expression differences, we assembled the transcriptome from the cDNA sequencing reads using the dedicated package Trinity. After the clustering of highly similar sequences (>95% homology), our identification yielded 783,260 transcripts or gene isoforms, which were further clustered into 548,894 genes by the Salmon version 1.6.0 software. After the identification of differentially expressed genes by the edgeR 3.14.0 software, which filtered out low expressed isoforms by excluding transcripts that had an average count lower than 2 or were counted only in a single sample, this number shrunk to 39,384 transcripts and 39,040 genes. These numbers are comparable to the previously published 61,994 transcripts and 42,478 genes in the *L. stagnalis* nervous system [12]. Out of these transcripts, 20,830 protein-coding transcripts had a significant homology to the 11,920 proteins in the Uniprot database; 20,499 had significant homology to the 10,411 human or mouse protein sequences; and 14,167 had significant homology to the 7537 characterized proteins of six mollusk species. Thus, on average, about two transcripts encode for a protein. The protein percent identity distributions were the same in the human–mouse and Uniprot databases (Appendix A). When compared to the search against selected six mollusk species as a reference, as expected, we observed a better homology to the characterized transcripts in mollusks, although we obtained less homologous transcripts, suggesting that a subset of the human–mouse or Uniprot transcripts are uncharacterized in mollusks (Appendix A). Therefore, to perform subsequent functional annotations, we used 20,137 transcripts without ribosomal internal transcribed spacer (ITS), encoding proteins homologous to entries in human or mouse databases (Appendix A).

### 2.2. Differential Gene Expression

The differential gene expression analysis identified induced and repressed genes after exercise and rest (Figure 2).

Interestingly, we identified a distinct subset of genes that are repressed after exercise and induced after rest (Figure 2A,B). The sequence analysis of these genes revealed the presence of homology to the ribosomal internal transcribed spacer—a poorly conserved region in between 18S, 5.8S, and 28S ribosomal subunits (Figure 2B). Taking into account that we depleted ribosomal RNA and used the poly-A sequence for reverse transcription, it is not clear at the moment what these transcripts represent. A total of 354 of these are protein-coding, and many of these are ribosomal proteins. The rest might represent a subset of the ribosomal RNA that were not sufficiently depleted and, in addition, contain poly-A-like sequences, as it was shown for human cells [13] and for yeast [14]. We attempted to validate the observed decrease in RNA after exercise followed by the increase in the rested animals for three of these transcripts and were able to reproduce it for two ribosomal transcripts and for GLUT1 transcript (Figure 3).

The validation of these transcripts confirmed patterns of gene expression changes for all tested transcripts, although it also confirmed a high variability of the data. Only the 1288 transcript had *p*.adj = 0.005 in comparison to exercise, while the others were not significant. And, *p* < 0.05 was reached only for the rest versus exercise comparisons (Figure 3). This was consistent with the FDR in the range of 0.13–0.25 for the analyzed transcripts in the edgeR analysis of the sequencing data.

Apparently, transcripts that were affected immediately after exercise became normalized after 2 h rest, whereas completely distinct transcripts were induced or repressed (Figure 2A,B). This suggests that gene expression changes are relatively fast. Therefore, when we select a stringent threshold, we will likely include in our analysis only transcripts that change slowly and, therefore, synchronize well, whereas genes whose transcription change up and down a little faster will vary more in between animals and would be missed from our analysis. Therefore, we chose to include transcripts with relaxed FDR and non-adjusted *p*-value < 0.01.

For the purpose of this paper, we excluded ITS-containing sequences (1447) from the annotation analysis and focused on the protein-coding transcripts lacking the ITS sequences (20,137 transcripts) (Figure 2C). In this group, 267 were induced by exercise and 308 transcripts were repressed (Figure 2C,D; FC > 2 or FC < 0.5, *p* < 0.01). After rest, we detected 83 induced transcripts and 205 repressed transcripts in the ganglia of rested animals (Figure 2C,D; FC > 2 or FC < 0.5, *p* < 0.01). Apparently, genes regulated after rest or exercise represent four distinct groups with little overlap, which is not significantly different from what we would expect by changes if data obtained after exercise or rest represent independent experiments (Figure 2D, *p* = 0.65, chi-squared test). There is a single gene that is induced by exercise and after rest (homologous to human Heat shock protein beta-8 associated with neuromuscular diseases) and three genes in the groups of down- or up-regulated transcripts in both conditions (Figure 2D).

### 2.3. Annotation of Genes That Change Their Expression After Exercise and Rest

We simultaneously characterized induced and repressed genes in response to exercise or rest by the Metascape 3.5 software [15] using human gene IDs for annotation (Figure 4). This analysis produces clusters of annotations using mixed gene ontology, KEGG, and Reactome annotations, in which the spot sizes reflect group sizes and the connecting lines reflect that five or more genes are common in between annotations according to the analysis settings.

Thereby, highly interconnected clusters have many genes in common (Figure 4, left side). The blue and red colors of the spots highlight a fraction of up or downregulated genes (Figure 4, right side). Notably, annotation clusters of transcripts with FDR < 0.05 include cell cycle, DNA metabolic processes, vesicle-mediated transport, and neuron morphogenesis up- or downregulated upon exercise or rest, similarly to the annotations of all transcripts (Figure 4, Appendix A). In addition, hypoxia-responsive genes were revealed by annotating exercise-induced transcripts with FDR < 0.05. Below, we will discuss the most interesting findings.

#### 2.3.1. Annotation of Genes Induced or Repressed upon Exercise

Exercise temporally induced or repressed a number of chromatin-modifying and transcriptional regulatory genes, which mostly were normalized after rest (Figure 5A).

**Figure 5 ijms-26-06970-f005:**
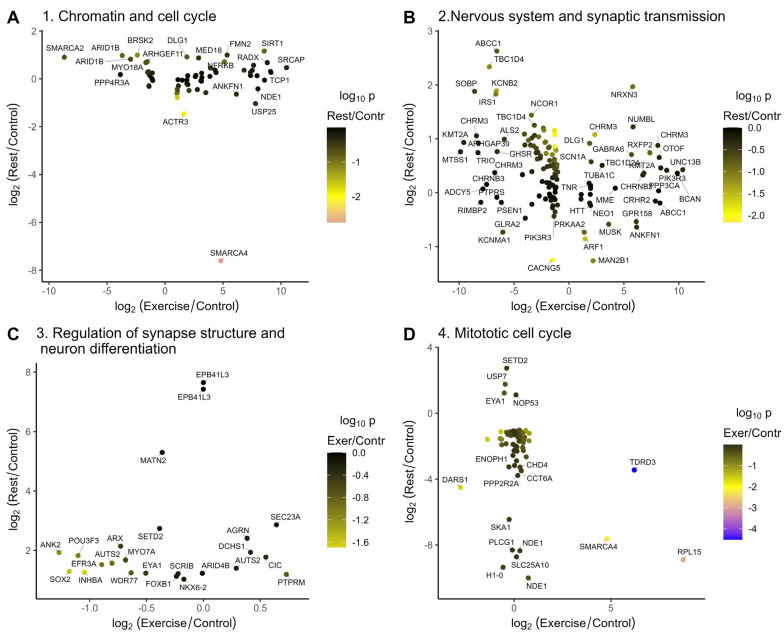
Genes from the annotation clusters that are misregulated after exercise or rest. The majority of transcripts that are repressed or induced upon exercise are normalized in the rested animals. Colors represent log_10_(*p*-values) for changes in gene expression in the rested relative to the control animals. (**A**) Changes in gene expression after rest versus exercise for the chromatin regulation and cell cycle related annotation clusters induced after exercise (Figure 4B, cluster 1); (**B**) Changes in gene expression after rest versus exercise for the nervous system and synaptic transmission related clusters repressed after exercise (Figure 4B, cluster 2); (**C**) Changes in gene expression after rest versus exercise for the neuron differntiation and synapse structure related clusters induced after rest (Figure 4D, cluster 3); (**D**) Changes in gene expression after rest versus exercise for the cell cycle related clusters repressed after rest (Figure 4D, cluster 4).

Among these, isoforms of SMARCA4, TDRD3, and C17orf49/BAP18 were induced upon exercise and were repressed after rest, and TASOR and BAZ2B were induced by the exercise and remained slightly elevated after rest, suggesting that they might function in a gene expression switch observed in our dataset. The TDRD3 is a part of the TOP3B complex that functions as a transcriptional coactivator and as a topoisomerase [16]. TDRD3 and TOP3B deficiency in mice led to the neuronal defects, including impairment in cognitive behavior, synaptic plasticity, and the downregulation of the mRNA associated with GABAergic neurons [17,18]. Brg1/SmarcA4 is a SWI/SNF-like ATP-dependent chromatin remodeling complex. It is pivotal for dendritic spine/synapse elimination and its mutations are associated with autism spectrum disorders [19,20]. The neuronal functions of the third gene induced by exercise and repressed by rest, C17orf49/BAP18, encoding a compound of MLL/NRF are unknown [21,22]. However, two transcripts encoding a Kmt2a/MLL1 H3K4 methyltransferase are induced and repressed by exercise and normalized after rest. Kmt2a depletion in the prefrontal cortex of adult mice resulted in decreased H3K4 methylation and impaired learning and emotions [23].

The chromatin remodeling and DNA/RNA-binding protein BAZ2B [24] is induced by exercise and remains slightly elevated after rest (FC = 1.88, *p* = 0.024). BAZ2B haploinsufficiency was identified in patients with intellectual disability and autism [25,26]. Another gene induced by exercise and whose level remains slightly elevated after rest is TASOR (FC = 2.77, *p* = 0.015). TASOR is a component of the RNA-guided HUSH transcriptional repressor complex depositing H3K9me3 at the non-self mobile genetic elements [27,28]. A lower H3K9me3 was observed in the protocadherin gene cluster in the brains of TASOR-knockout mice who have a decreased lifespan and motor activities and increased fear memory [29].

The most induced gene from the chromatin remodeling and gene regulation annotation cluster is SRCAP, a transcriptional activator and chromatin remodeling ATPase that inserts H2A.Z into chromatin and whose mutations are associated with neurological syndromes [30,31].

Exercise-repressed TRRAP encodes a histone acetyltransferase cofactor [32] functioning in the DNA DSB repair [33]. The mutations of TRRAP are recently associated with multiple abnormalities and characterized by intellectual disability or autism spectrum disorders [34]. Interestingly, TRRAP is required for SP1 binding at the promoters of tubulin dynamics genes in Purkinje neurons, and neuronal defects are rescued by Stathmin 3/4 overexpression [35].

Among other exercise-repressed genes from this cluster, ten–eleven translocation methylcytosine dioxygenase 2 (Tet2) catalyzes the oxidation of 5-methylcytosine to 5-hydroxymethylcytosine. Tet2 expression is restored after rest with the level slightly exceeding the control (FC = 2.2, *p* = 0.035). Interestingly, enhanced cognitive functions were reported in mice with Tet2 deficiency [36], while other data suggest the opposite [37].

Similarly, exercise-repressed ARID1B (that is also upregulated or downregulated in different hippocamp neurons of exercised mice, Table 1) is a component of the Smarca4/Brg1 remodeling complex [38]. ARID1B is involved in GABA neuron development, and mice deficient in ARID1B serve as a model of autism [39,40,41]. Exercise-repressed NCOR1 also regulates GABAA receptor subunit α2 expression and memory formation [42].

Thus, exercise and subsequent rest in *L. stagnalis* regulate multiple transcriptional activators and chromatin-remodeling genes, which might be involved in the gene expression switch and are involved in the pivotal neuronal and cognitive functions in other species.

The major annotation clusters downregulated after exercise are behavioral, neuronal system, and synaptic transmission gene clusters (Figure 4B, cluster 2, Figure 5B). Consistent with the increase in serotonergic neuronal activity during exercise (which is under the negative control of dopamine in *L. stagnalis*), the most repressed genes in this cluster (Figure 5B) include genes functioning in dopamine signaling: ADCY5 [43], CASK [44], RIMS1 [45,46], and CPLX1 [47]. In addition, genes involved in learning and memory (ARHGAP39 [48] and HERC1 [49]), intellectual disability (SOBP [50]), or autism (NF1), whose mutations cause neurodevelopmental, behavioral, learning, and motor abnormalities [51,52], are repressed by exercise and normalized or slightly induced upon subsequent rest (Figure 5B). Interestingly, five of these genes are reported to be involved in response to exercise in the hippocamp in mice [53], namely CASK (downregulated in mice and in snails), RIMS1 (up or down in different neurons of mice, repressed in snails), ARHGAP39 (upregulated in mice and in snails after rest), and HERC1 and SOBP (upregulated in mice, down–upregulated in snails).

Hormone response genes, which are repressed by exercise and normalized after rest, are involved in pivotal neuronal functions (Appendix A). Among these are the acetylcholine receptor CHRM3, which is presented by different isoforms in snail CNS. CHRM3 is involved in sleep regulation [54]; GHSR in regulating feeding behavior, addiction, and dopamine signaling [55,56]; and IRS1 in regulating neurite elongation and body growth [57].

Vesicle-mediated transport, targeting, and localization genes are induced upon exercise and repressed after rest (Appendix A). Within this group, ARF1 is involved in *C. elegans* via Ca^2+^ sensing [58], STON2 is pivotal for synaptic vesicle protein sorting [59], MGAT1 functions in Drosophila brain development and locomotion [60,61], and TBC1D24 is involved in neuronal migration, maturation, and neurotransmission [62,63].

#### 2.3.2. Annotation of Genes Induced or Repressed upon Rest

The heat shock protein beta-8 HSPB8 (HSPB11 or HSPB22 [64]) is a single gene that is significantly induced by exercise and remains significantly induced after rest (Figure 2D). HSPB8 is a core component of the Chaperone-Assisted Selective Autophagy (CASA) complex and counteracts misfolded protein accumulation in neurodegenerative diseases [65,66,67,68]. The mutation of the HSPB22 causes distal motor neuropathy [69].

The most notable gene annotation cluster induced upon rest is a cluster of neurodevelopmental, synapse structure, and neuron differentiation (Figure 4D, cluster 3, Figure 5C). In this rest-induced group, a few genes were repressed by exercise, including SOX2 and ANK2, which is up- and downregulated in different neurons of exercised mice [53], and POU3F3 and KDM6A, which are required for neurodevelopment and learning [70,71,72,73,74,75]. Likewise, the transcriptional activator AUTS2 [76,77], KCNH7 potassium channel variants [78,79], lysine methyltransferase SETD2 (which is also downregulated in mice) [53,80,81], and USP7 [82,83], among others, are induced by rest, and their mutations are associated with a variety of neurodevelopmental disorders.

The ROBO2 axon guidance receptor transcript is induced by rest and elevated right after exercise (FC = 5.3, *p* = 0.025), functioning in the inhibition of dopamine signaling and neuron projection development [84,85,86]. In turn, the rest-induced E3 ubiquitin ligase MYCBP2 reportedly cooperates with ROBO2 in the olfactory system development in mice and is associated with neurodevelopmental problems and autism in humans [87,88].

Rest-induced INHBA is a BDNF calcium-induced transcriptional effector that suppresses NMDA-mediated toxicity [89]. EFR3A is induced by rest in a snail and by four weeks of voluntary exercise in the hippocamp of mice [53]. The protein controls G protein-coupled receptor (GPCR) activity by affecting receptor phosphorylation. Whole-exome sequencing studies have implicated mutations in this gene with autism spectrum disorders [90]. In contrast, other studies suggest that its loss is neuroprotective [91]. Among other rest-induced genes, A-kinase anchor protein-encoding AKAP13 binds to estrogen receptor, thereby promoting estrogen action in the brain, and its haploinsufficiency causes obsessive behavior in mice [92].

The clusters of genes annotated as associated with mitotic cell cycle or chromatid separation are mostly repressed by rest (Figure 4C, cluster 4, Figure 5D). One of the most repressed genes is NDE1, which is essential for progenitor cell proliferation and neuronal migration [93] and whose mutations are associated with schizophrenia [94]. We found the repression of phospholipase C, which is also repressed in mice after exercise and whose deletion is associated with enhanced dopamine release in mice [53,95].

In addition, the transcriptional program of *L. stagnalis* neurons in response to rest is characterized by the repression of genes involved in DNA damage response, apoptosis, translational regulation, and protein ubiquitination (Figure 6).

Among them, protein kinase DYRK1A is associated with severe neurodevelopment disorder [96]. DYRK1A interacts with E3 ubiquitin ligases and transcriptional elongation factors, while low levels of DYRK1A might promote cell survival in response to genotoxic stress [97].

Accordingly, apoptosis might be inhibited by the lower IPTR3 expression, which is regulated by SMARCA4 and controls Ca^2+^ flux in the mitochondria [98,99]. Rest-repressed GADD45G is a tumor suppressor, known to be involved in the induction of apoptosis in multiple cell types [100], and in the neurons, it is required for memory consolidation by promoter demethylation [101,102] and for neurite outgrowth [103].

In contrast, low levels of BIRC2, XIAP, and EIF2S1 might promote cell death [104,105,106].

Exercise-induced and rest-repressed disease-associated O-GlcNAc transferase, OGT, is involved in gene regulation by interaction with TRIM28 and TET2 and glycosylation of chromatin and transcriptional regulators [107,108,109,110].

Notably, decreased O-GlcNAc promoted the binding of the E3-ubiquitin ligase and Notch intracellular domain, causing NICD degradation, the inhibition of Notch signaling, and depletion of the neuronal stem cells in mice [110].

The functions of Desumoylating isopeptidase 1 (DESI1) in neuronal cells have not been investigated.

In contrast, rest-induced USP19 de-ubiquitinates proteins, while rest-induced probable E3 ubiquitin-protein ligase MYCBP2 likely ubiquitinates proteins. Together with changes in other transcripts encoding proteins involved in protein synthesis, modification, and catabolism, these data suggest that rest leads to selective changes in protein turnover and activity.

Rest represses the expression of several translation-related proteins, including DDX56, which is involved in the expression of ribosomal RNAs in stem cells [111], and Peptidyl-tRNA hydrolase MRPL58 also known as ICT1, a mitochondrial ribosomal protein, functioning in translation termination and also involved in the suppression of apoptosis [112]. Rest also downregulates Nip7, which is required for 18S rRNA maturation [113]. In addition, it was recently reported that the depletion of the above-mentioned rest-repressed DYRK1A causes a general reduction in the ribosomal proteins [114]. Altogether, this suggests that protein translation is likely repressed after rest.

Thus, rest normalizes the exercise-misregulated genes and induces the expression of other genes functioning in transcription, dopamine signaling, neuron differentiation, and functions. Homologous genes are pivotal for normal development and complex neurological functions and their mutations are associated with behavioral, learning, and neurodevelopmental abnormalities in model animals and humans. The analysis of the literature revealed that, surprisingly, many genes homologous to those affected by exercise and rest in *L. stagnalis* are implicated in the pathophysiology of autism spectrum disorders and other neurodevelopmental and neurodegenerative diseases [19,20,76,77].

### 2.4. Genes That Are Similarly Regulated by Exercise in Snails and Mice Revealed Conserved Functional Annotations

We evaluated if the overlap of genes misregulated in snails and in mouse hippocampus after voluntary exercise differs from a random one [53]. There are 9806 genes in snails that are homologous to mice and can be mapped to mouse ETEREZID. There are 328 induced and 811 genes repressed by exercise in the mouse hippocampus. Out of 314 transcripts repressed by exercise in a snail with unique mouse annotations, 50 were downregulated and 25 up-regulated in mice (26 and 11 expected by chance, *p* = 4 × 10^−10^, Pearson’s chi-squared test), whereas out of 252 transcripts induced by exercise, 31 were downregulated and 9 were induced (21 and 8.5 expected by chance, *p* = 0.06, Pearson’s chi-squared test) in mouse hippocampus.

Out of 196 transcripts repressed by rest with unique mouse annotations, 14 were downregulated and 2 upregulated in mice (16 and 7 expected by chance, *p* = 0.14), whereas out of 85 transcripts induced by rest, 13 were downregulated and 8 were induced (7 and 3 expected by chance, *p* = 0.06, 7.6 × 10^−4^ Fisher’s exact test) in mouse hippocampus.

Thus, a subset of transcripts repressed by exercise and induced by rest in snails are also misregulated in mice with higher than expected frequencies (Appendix A).

Table 1 shows 20 the most significantly changed genes regulated by exercise or rest both in snails and mice.

**Table 1 ijms-26-06970-t001:** The most significantly changed genes that are regulated by exercise or rest in snails and by voluntary exercise in mice [52]. DEG is differentia gene expression.

RefSeq Accession Mber	SYMBOL	log_2_FC Mouse	p_adj_ Mouse	log_2_FC Snail	FDR Snail	DEG Mouse	DEG Snail
NP_032939.1	Ppp3ca	−90.6	3.65 × 10^−4^	8.2	5.85 × 10^−6^	Down	UP Exer
NP_064450.3	Fmn2	−5.9	0.0126	3.9	0.000507	Down	UP Exer
NP_032939.1	Ppp3ca	−90.6	3.65 × 10^−4^	2.0	0.002533	Down	UP Exer
NP_001078824.1	Arid1b	−0.2	0.04065	−3.7	0.002018	Down	Down Exer
NP_055726.4	Aak1	−7.0	1.59 × 10^−8^	−3.8	2.59 × 10^−5^	Down	Down Exer
NP_055726.4	Aak1	−7.5	0.004973	−3.8	2.59 × 10^−5^	Down	Down Exer
NP_084542.2	Inpp4a	−4.6	6.46 × 10^−4^	−5.3	4.17 × 10^−7^	Down	Down Exer
NP_001366215.1	Zfp423	−8.0	4.19 × 10^−11^	−9.8	0.001463	Down	Down Exer
NP_001078824.1	Arid1b	4.6	2.24 × 10^−5^	−3.7	0.002018	UP	Down Exer
NP_663592.3	Herc1	13.7	1.62 × 10^−4^	−10.9	3.54 × 10^−15^	UP	Down Exer
NP_899233.1	Asic2	−62.4	5.07 × 10^−4^	9.1	0.000237	Down	UP Rest
NP_036439.2	Epb41l3	−19.9	2.42 × 10^−10^	7.7	0.014063	Down	UP Rest
NP_036439.2	Epb41l3	−19.9	2.42 × 10^−10^	7.4	6.6 × 10^−5^	Down	UP Rest
NP_055847.1	Pds5b	−6.7	0.026451	2.1	0.001158	Down	UP Rest
NP_001103785.1	Kif1a	−4.5	1.65 × 10^−6^	1.5	0.028579	Down	UP Rest
NP_001161760.2	Arhgap39	4.6	5.13 × 10^−4^	7.6	0.037305	UP	UP Rest
NP_055952.2	Efr3a	16.1	4.22 × 10^−6^	1.5	0.01432	UP	UP Rest
NP_058616.1	Atp6v0a1	−10.9	0.017536	−1.4	0.040801	Down	Down Rest
NP_085098.1	Nbea	−36.4	9.15 × 10^−5^	−10.1	8.51 × 10^−7^	Down	Down Rest
NP_004761.2	Kcnb2	60.9	0.001578	−9.1	0.000163	UP	Down Rest

The analysis of common exercise regulated genes by Metascape revealed synaptic signaling, chemical synaptic transmission, and autism spectrum disease-related annotations of human homologous genes (Figure 7).

ARRDC2 and Unc80 are among the genes upregulated by exercise in both snails and mice. ARRDC2 protein is located in cytoplasmic and vesicle membranes, and belongs to the arrestin protein family that plays an important role in the desensitization and internalization of G protein-coupled receptors (GPCRs) [115,116]. In snails, it remains slightly upregulated after rest as well. Acute aerobic exercise increased Arrdc2 and Arrdc3 expression also in skeletal muscle [117], and Arrdc2 was suggested to be involved in disuse atrophy, particularly in aged muscles [118].

Unc80 isoforms are up- and downregulated in both snails and mice. Unc80 encodes a protein that is a component of the voltage-independent “leak” ion-channel complex conserved in invertebrates and mammals [119]. Leak channels play an important role in the establishment and maintenance of resting membrane potentials (MPs) in neurons. At least in snails, changes in MP were indeed detected in neurons after exercise in CNS and after complete isolation [11]. It is likely that long-term changes induced by voluntary exercise are associated with MP regulation in some hippocampal neurons in mice as well. However, it is still problematic to detect MP tuning in neurons in a mammalian brain. Notably, Unc 80 is widely distributed in the nervous system of *C. elegance*, and Unc80-knockout nematodes, being able to crawl, are unable to switch to and perform more intense and fast locomotion and swimming exercise [119]. Therefore, Unc 80 is indeed involved in the neuronal and neuromuscular background of exercise in various species. Additionally, mutations in Unc80 are associated with congenital infantile encephalopathy, intellectual disability, and growth issues [120].

Among transcripts downregulated in snail CNS and mouse hippocampus by exercise are Inpp4a; Aak1; Arid1b; Smurf2; Bptf; Dgkb; Mllt10; and Cacnb2.

Inositol Polyphosphate-4-Phosphatase Type I A (Inpp4a) modulates cell cycle progression and cell survival. In neurons, it is involved in the regulation of vesicle transport, cytoskeletal reorganization, and expression of N-methyl-D-aspartate-type glutamate receptors (NMDARs) at the cell surface, protecting neurons from excitotoxicity and death [121].

Aak1 belongs to the SNF1 subfamily of serine/threonine protein kinases and is also involved in receptor endocytosis via the clathrin axis and migration of neuroblast daughter cells [122]. The downregulation of its function helps to reduce pain [123], while increased content is associated recently with major depressive disorder [124]. These findings agree with the observed downregulation of AAK1 after exercise and known antinociceptive and antidepressive effects of exercise.

The ARID1B gene product is also involved in the proliferation and differentiation of neural precursors by chromatin remodeling and repairing damaged DNA. Together with SMARCA4, it belongs to the neural progenitor-specific chromatin-remodeling complex (npBAF complex) and the neuron-specific chromatin-remodeling complex (nBAF complex), which causes a switch from a stem/progenitor to a postmitotic chromatin-remodeling mechanism [125,126,127]. ARID1B is one of the most frequently mutated genes in intellectual disability cohorts [128].

Mllt10 and Bptf are transcriptional factors. Bptf (Bromodomain And PHD Domain Transcription Factor) is also involved in chromatin remodeling and is one of the crucial regulators of neurogenesis pathways within the NURF complex together with above-mentioned SMARCA4, similarly dysregulated by exercise in snails. The downregulation of Smurf2 might manifest the upregulation of a highly conserved signaling pathway, namely transforming growth factor beta receptor (TGFBR). TGFBR is necessary for neural development and nervous system function throughout life and is dysregulated in neurodegenerative diseases [129].

The downregulation of Dgkb transcripts (Diacylglycerol kinase beta) in both snail and mice suggests an increased production of diacylglycerol/DAG and a decrease in the production of phosphatidic acid after exercise. Dgkb converts diacylglycerol/DAG into phosphatidic acid/phosphatidate/PA and regulates the respective levels of these two bioactive lipids. Again, this finding suggests changes in the GPCR and elevated cytosolic Ca^2+^ signaling. Interestingly, Dgkb KO mice have several psychomotor behavioral changes, such as reduced anxiety and depression, and hyperactivity [130]. These changes are similar to the behavioral consequences of exercise in mice and in snails. It is thus tempting to speculate that, at least partially, these behavioral effects may be mediated by a decreased number of Dgkb transcripts after exercise. DGKβ was also shown to regulate spine formation in dendrites, playing an important role in cognitive processes including memory [131].

The analysis of a few common rest-regulated genes in snails and exercise-regulated genes in mice by Metascape revealed cell junction organization genes and a few putative protein interactions.

Thus, it was reported that rest-repressed (in snails) and exercise-repressed (in mice) DYRK1A, O-GlcNAc transferase [132], and Clathrin [133] and rest-induced (down in mice) LRBA [97] all interact with DYRK1A. In turn, rest-induced (down in mice) ARID4B might interact with rest-induced BCL11A (up in mice); both are associated with neurodevelopmental disorders [134,135].

## 3. Discussion

An intense crawling exercise is a model of *L. stagnalis* behavioral adaptation to changes in the environment, specifically shallow water. The primary aim of our investigation was to identify genes whose expression may relate to the adaptive changes in the CNS, produced by sudden shallow water crawling (exercise) and a subsequent return to deep water (rest) in *L. stagnalis*.

Two hours of exercise in low water has previously been reported to produce immediate and postponed behavioral changes, such as increased reproduction [136], higher behavioral activity, and to facilitate decision-making in a new completely dry arena [8,10,11]. Exercise was accompanied by the induction of serotonergic neuronal activity followed by dopamine-dependent repression during rest [11]. The data obtained in our previous behavioral, chemical, and electrophysiological experiments suggest that rest following exercise is a specific state of an organism that differs from both the exercised and the non-trained controls [10,11]. The results of transcriptome analysis agree with these observations.

### 3.1. Exercise and Rest After Exercise Cause Remarkably Different Gene Expression Programs

Accordingly, in this paper, we show that nearly all genes misregulated after 2 h of exercise were normalized after 2 h of rest following exercise, which, in turn, caused the misregulation of the distinct gene set.

These data suggest that rest does not manifest itself as an intermediate state between exercise and non-trained control at the gene expression level. In contrast, at two hours post-exercise, rest represents a physiologically distinct state. Although, after a certain time, the exercised snails may return to their untrained state or control level, it is clear that the trajectory of this return is complex and different from the trajectory of transition to the exercised state.

The only transcript whose expression was upregulated by exercise and remained upregulated in rest is homologous to human Heat shock protein beta-8 HSPB8 (alternatively called HSPB11 or HSPB22 [64]). In neurons, HSPB8 has been demonstrated to protect against neurotoxicity in several models of neurodegenerative diseases, such as amyotrophic lateral sclerosis and fronto-lateral temporal dementia, by facilitating autophagy [66,67,137]. This gene is of clinical importance in humans, as its product plays a protective role in a number of neurodegenerative diseases that are associated with misfolded protein accumulation [65,68,69].

Although it remains unclear if the *L. stagnalis* transcript is a true ortholog of HSPB8, our finding is the first evidence that HSPB8 is activated by exercise, which is well-known to protect from neurodegeneration.

### 3.2. Down- and Up-Regulation of Ribosomal Genes by Exercise and Rest: Unexpected Findings of Differential Analysis

Unexpectedly, a distinct subset of genes with homology to the ribosomal internal transcribed spacer was repressed after exercise and normalized after rest. In turn, a distinct set of transcripts encoding proteins involved in translational regulation were repressed following rest. The functional significance of this response is yet to be understood. It is known that an eukaryotic cell is able to quickly respond to a sudden stress by the rapid suppression of protein synthesis [138,139]. Ribosomal protein biosynthesis is regulated by the nutrient availability in distant organisms. For instance, the conserved rapamycin (TOR) signal transduction pathway adjusts the ribosomal protein biosynthetic capacity to nutrient availability in yeasts and mammals [138].

In neurons, a number of mechanisms modulate ribosomal functions [140]. For example, changes in the local ribosomal activity in distal cellular locations allows the remodeling of the local proteome, underlying dendritic changes and neuronal plasticity [140].

During periods of high-energy release like exercise, neurons may try to stop additional energy-consuming processes, including protein synthesis. The rebound effect observed after the exercise agrees with this suggestion. Interestingly, more than 50 years ago, an inhibition of mRNA synthesis followed by rebound was seen in response to the electric stimulation of neurons in *L. stagnalis* [141]. The genes encoding ribosomal proteins were downregulated by the incubation of a hippocampal neuronal culture in picrotoxin, which causes strong excitation [142]. The authors suggested that this may indicate the redistribution of transcriptional resources. Another possibility is that, during periods of high risk for DNA stability, the decrease in DNA transcription and RNA translation may help to protect neuronal DNA [143]. The same may explain the downregulation of translation-related genes upon rest.

### 3.3. Rest After Exercise Is Important for Neurodevelopmental Gene Activation Across Species

In vertebrates, exercise is known to produce time-dependent consequences of various neuromodulatory events (including serotonin and dopamine transmission), many of which are favorable for brain functions [3]. Additionally, exercise is known to act on epigenetics and chromatin organization in the hypothalamus and frontal lobes in rodents [5]. The general shift to more open chromatin states resembles the effect produced by novelty [144] and agrees with the idea that this shift facilitates learning and memory [145,146]. Indeed, *L. stagnalis* exercise regulates multiple transcriptional co-activators and repressors, while rest induces another set of transcriptional modulators that likely mediate the switch between two distinct gene sets affected by exercise and rest. Remarkably, deficiencies of these transcriptional regulators are implicated in neurodevelopmental defects in mice and humans, including TDRD3 [17,18], SmarcA4 [19,20], Kmt2a/Mll1 [23], TRRAP [34], and AUTS2 transcription factor [76,77], among many others, suggesting that these proteins are pivotal for the neuronal functions and likely impact memory and decision-making in *L. stagnalis*. The functions of these transcripts in *L. stagnalis* require further experimental validation.

Consistent with the dopamine-mediated repression of neuronal activity, genes related to dopamine signaling, namely KLRN [147], ROBO2 [84], ADCY5 [43], CASK [44], RIMS1 [45,46], and CPLX1 [47], are repressed by exercise and induced by subsequent rest. In addition, genes involved in learning and memory are restored after rest after being repressed by exercise, including GABARB1 [148], curiosity NRCAM [149], and GABA signaling RBPJ [150] and ARID1B [40,41].

Even a superficial analysis of the functions of genes that changed activity after exercise in mice and snails shows that these changes are certainly relevant to neurological functions, suggesting that the transcriptional mechanisms behind the beneficial effects of rest might be conserved across species. Future studies on a wider spectrum of species will help to elucidate whether these genes indeed belong to the most conserved taxa targets of physical exercise.

To summarize, these findings point to rest after exercise being the most promising state for the elucidation of the mechanisms associated with cognitive function activation.

## 4. Methods

### 4.1. L. stagnalis Rest and Exercise Trials

We used a previously developed model for simulating intense locomotion in *L. stagnalis* by decreasing the level of water in a container with snails [8,10,11] (Figure 1). Under these conditions, the so-called terrestrial form of muscle crawling is activated in the pond snail, which is energetically more expensive and rarely used by the mollusk in normal aquarium conditions. The body of the animal is hydrated, preventing it from drying out, and at the same time, the snail is forced to crawl, using intense muscle contraction to compensate for the lack of a water column supporting the shell [8].

Twelve groups of six animals each were used: three groups of active locomotion (exercise, E), three groups of active locomotion followed by rest (exercise–rest, ER), and six control groups (C), which were used for the E and ER groups, respectively. The snails of the E groups were placed in a container with a bottom area of 50 × 50 cm, filled with a thin (2 mm) layer of water, for 2 h. The snails of the ER groups were treated as above and then returned to the aquariums for 2 h. The control groups were kept in aquariums with a sufficient layer of water that does not change the locomotion regime compared to the usual conditions and were processed at the same time as the corresponding experimental animals. The central ganglia from each group of animals were dissected under 0.1 M MgCl_2_ anesthesia (0.5 mL per animal for 1 min) immediately after the end of the behavioral procedures and pooled for RNA extraction.

### 4.2. RNA Extraction, Library Preparation, and Sequencing

RNA was extracted using the ExtractRNA kit (Evrogen, Moscow, Russia), and the RNA integrity index was more than 7, as estimated by the Agilent 2100 Bioanalyzer (Agilent, Santa Clara, CA, USA). Ribosomal RNA was depleted by the MGIEasy rRNA Depletion Kit (MGI Tech Co., Shenzhen, China), followed by library preparation using the MGIEasy RNA Library Prep Set (MGI Tech Co., China). cDNA quality was estimated by the Agilent 2100 Bioanalyzer, NanoDrop 2000 (Thermo Fisher Scientific, Waltham, MA, USA), and electrophoresis. The sequencing was performed by MGISEQ-2000 Kit (MGI Tech Co., China). Raw sequencing data were deposited to the SRA archive study SRP420638, which is associated with the bioproject PRJNA924952.

### 4.3. RT-PCR

RNA was reverse-transcribed following the recommendations of Evrogen RT-RCR kit (Evrogen, Moscow, Russia) using 500 ng of total RNA and oligo dT primers. Subsequently, cDNA was quantified by the real-time PCR using Evrogen SybrGreen mastermix, and the ΔCt method was used to estimate the quantities of the tested genes relative to the reference gene elongation factor 1-alpha (*EF1a*) [151]. Primers for glucose transporter 1 (*Glut1*) were TCAACGAACAAGGCCACAGA and AACGGCTTGCCATCTCGTAT, and, for *EF1a*, ACCACAACTGGCCACTTGATC and CCATCTCTTGGGCCTCTTTCT.

Primes for the ribosomal transcripts were generated by the Trinity 2.8.6 software.
TRINITY_DN87174_c0_g1_i3:
CCATCATTCCATGCACAATC,GGAGTTTGACTGGGGTGGTA;TRINITY_DN1288_c1_g1_i2:
CAGTGAGCTGAACCAGGACA, CACCACTTTTTGGCTGGATT:TRINITY_DN55636_c0_g1_i5:
TGATAGCTCCCCCTCGAATA,CGAGATTCCCACTGTCCCTA.

### 4.4. Transcriptome Assembly and Differential Gene Expression

#### 4.4.1. Transcriptome Assembly

The transcriptome assembly and data analysis was performed using Trinity 2.8.6 package [152]. Briefly, the analysis included the following steps:

Low-quality sequences were filtered out by the Trimmomatic software, which is a part of the Trinity 2.8.6., leaving in more than 99.6% sequences. Furthermore, to reduce the uncertainty of the subsequent gene expression profiling, transcripts were grouped by the cd-hit program [153], combining transcripts that were more than 95% identical. Thereby, the quantity of the transcripts was reduced from 964,489 to 783,260 (about 19% reduction). Using the transdecoder software (a part of Trinity 2.8.6.), it was possible to identify 242,579 protein-coding peptides in these data.

#### 4.4.2. Annotations of Transcripts by Known Homologous Sequences

Furthermore, homologous sequences were searched by the protein blastp BLAST 2.13.0 algorithm with -evalue 1 × 10^−4^ and reporting the first 5 sequences that satisfy the desired evalue. For annotation, we used either combined NCBI RefSeq proteins for *Homo sapiens* (taxid: 9606) and *Mus musculus* (taxid: 10090) sequences, Uniprot protein sequences, or combined Uniprot protein sequences for *Lottia gigantea* (taxid: 225164), *Elysia chlorotica* (taxid: 188477), *Mizuhopecten yessoensis* (6573), *Crassostrea gigas* (29159), *Mytilus coruscus* (42192), and *Pomacea canaliculata* (400727).

#### 4.4.3. Quantification of Differential Transcript Expression

The levels of transcripts were estimated inside the Trinity package by the salmon program with the weighted trimmed mean of the log expression ratios (TMM method) [154].

Differential transcript expression levels were estimated by the edgeR 3.14.0 package. The statistical differences between gene expression was calculated by the exact tests for differences in the means between two groups of negative-binomially distributed counts (the exactTest function). Transcripts with fold-change expression ratios in different conditions more than 2 with *p* < 0.01 were considered as differentially expressed.

#### 4.4.4. Gene Ontology and Pathway Annotations of Differentially Expressed Genes

Gene ontology and pathway enrichment of differentially expressed genes (up- and downregulated simultaneously) were determined by the Metascape 3.5 software with the default parameters, except that the minimum overlap between annotations to be connected with an edge was 5, instead of the default 3, and the selective clusters were chosen to be preferentially picked up by the algorithm. The names of the most significant annotations for a particular cluster along with a few other interesting significant annotations were manually added to Figure 4.

## 5. Conclusions

In this paper, we identified genes involved in the transcriptional response of *L. stagnalis* to shallow water intense crawling exercise and subsequent rest in deep water. This transcriptional response suggests that rest after exercise is functionally significant for the beneficial effects of exercise. Rest activates neurodevelopmental genes that may facilitate learning, memory, and environmental adaptation. We also observed similarities in the transcriptional response to exercise between rodents and mollusks at the level of large functional clusters and individual genes.

## Figures and Tables

**Figure 1 ijms-26-06970-f001:**
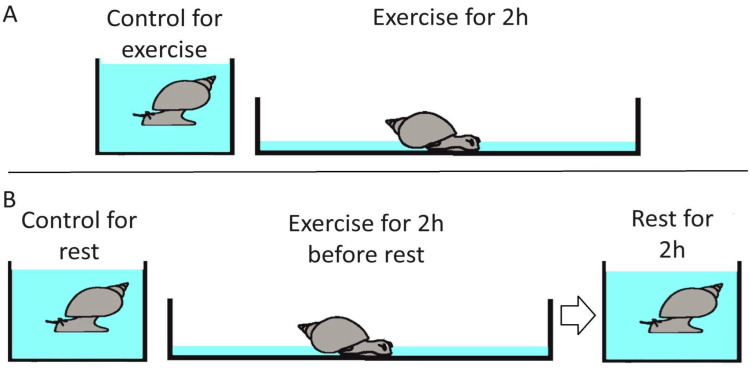
Experimental setup. Two types of experiments were performed: exercise and exercise followed by rest. (**A**) Exercise trails. The snail is forced to crawl for 2 h using intense muscle contraction to compensate for the lack of a water column supporting the shell. (**B**) Exercise followed by rest trails. The snail is forced to crawl for 2 h using intense muscle contraction to compensate for the lack of a water column supporting the shell. This was followed by rest in the normal aquarium conditions for 2 h. Mock-handled animals were kept in the normal aquarium conditions for 2 or 4 h.

**Figure 2 ijms-26-06970-f002:**
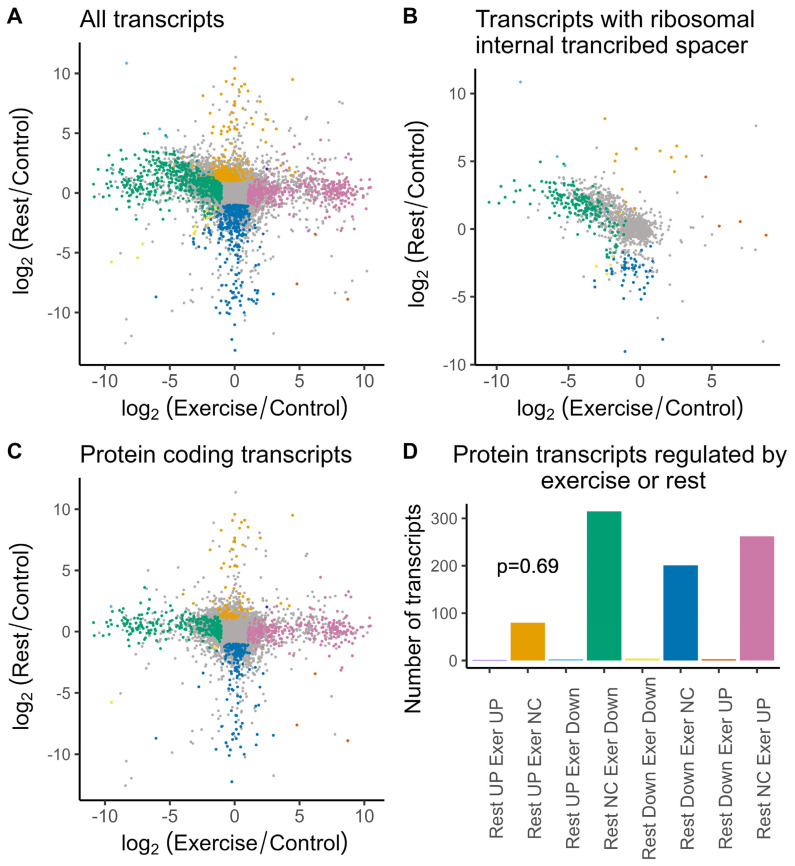
Genes differentially expressed in *L. stagnalis* in response to exercise and exercise–rest represent distinct groups. In colors are genes regulated by exercise and exercise–rest (FC > 2 or FC < 0.5, *p* < 0.01) and in gray are genes whose changes are not significant. (**A**) Scatterplot of binary logarithm for fold changes of isoform expression levels after rest versus exercise relative to the control values for all transcripts. (**B**) The same data plotted only for transcripts containing ribosomal internal transcribed spacer (ITS). (**C**) The same data as in (**A**), plotted for the putative protein-coding transcripts without ITS. (**D**) Number of protein-coding transcripts in groups commonly or differentially regulated by exercise or exercise–rest. NC stands for “no changes”. The intersections between groups were random according to the chi-squared test (*p* = 0.65).

**Figure 3 ijms-26-06970-f003:**
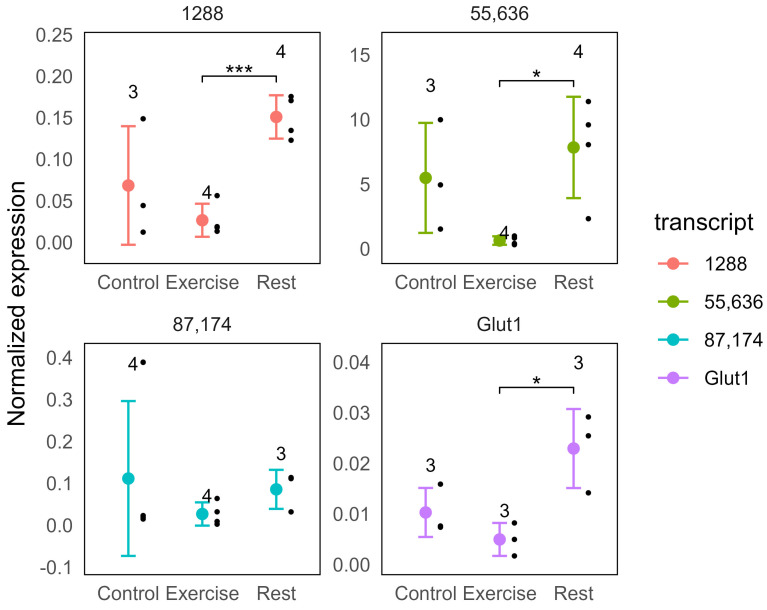
Changes in the normalized expression for selected ribosomal transcripts and transcript homologous to Glut1. The data are normalized to a reference gene EF1a. Colored circles are mean values, bars are standard deviations, and black dots individual animals. Stars highlight significant changes (* *p* < 0.05, *** *p* < 0.001, *t*-test), and numbers are the number of animals in groups.

**Figure 4 ijms-26-06970-f004:**
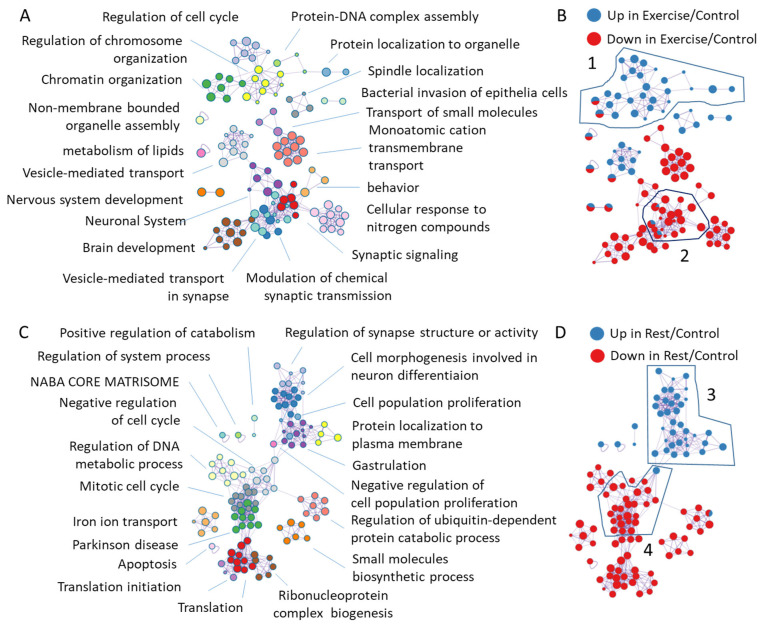
Clustering of genes misregulated by exercise or rest by the functional properties of the human homologous genes revealed the overrepresentation of annotations related to neuronal, developmental, and disease-related gene clusters. Dots represent annotation groups that are connected if there are 5 or more genes in common [15]. Annotation clusters are labelled by the most significantly overrepresented annotation. (**A**,**B**) Annotations of genes up- or downregulated by exercise relative to control revealed clusters of repressed genes related to ion transport, vesicle synaptic transport, neuronal system, and brain development, whereas annotation related to chromatin organization and cell division were induced. Other gene clusters contained both induced and repressed genes, including vesicle-mediated transport and nervous system development. (**C**,**D**) Annotations of genes up. or downregulated by rest relative to control revealed clusters of repressed genes related to translation, protein ubiquitination, mitotic cell cycle, Parkinson disease and apoptosis, whereas synapse assembly and neuron differentiation related genes were induced. Numbers on the right panels represent clusters annotated in Figure 5.

**Figure 6 ijms-26-06970-f006:**
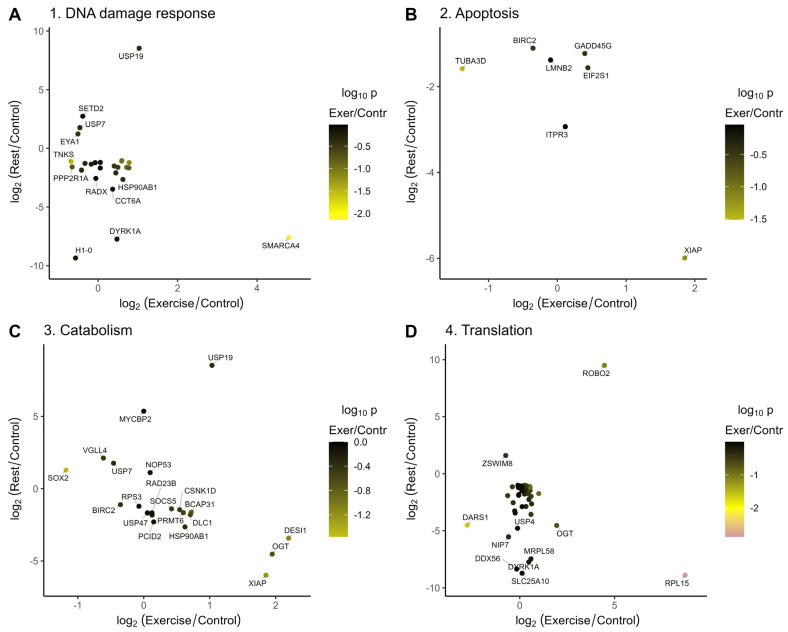
Annotation clusters of rest-regulated genes. The majority of transcripts that are repressed or induced upon exercise are normalized in the rested animals, while rest regulates distinct transcripts. Colors represent log_10_(*p*-values) for changes in gene expression in the rested relative to the control animals. (**A**) Changes in gene expression after rest versus exercise for the DNA damage response related annotation cluster repressed after rest (regulation of DNA metabolic process; Figure 4C); (**B**) Changes in gene expression after rest versus exercise for the apoptosis cluster repressed after rest (Figure 4C); (**C**) Changes in gene expression after rest versus exercise for the ubiquitin dependent protein catabolism cluster repressed after rest (Figure 4C); (**D**) Changes in gene expression after rest versus exercise for the translation related genes repressed after rest (Figure 4C).

**Figure 7 ijms-26-06970-f007:**
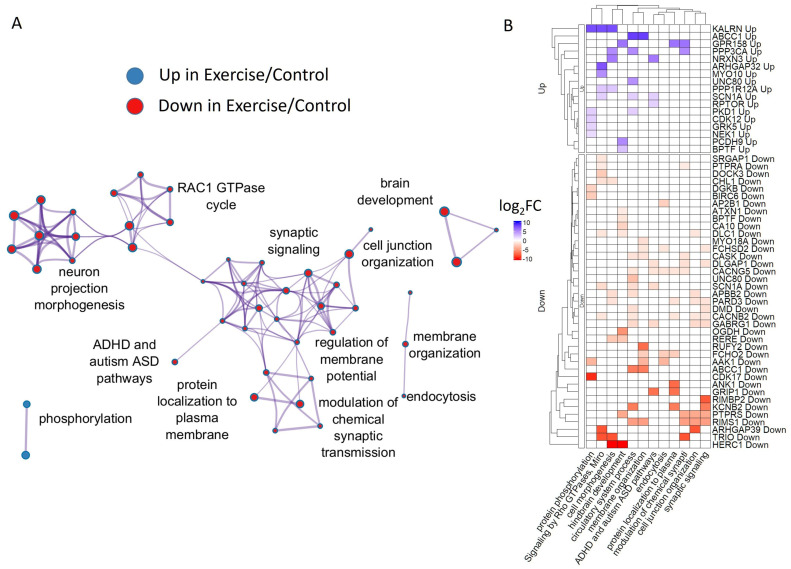
Annotation of genes regulated by exercise in snails and in mice. (**A**) Clustering of annotations overrepresented in the lists of misregulated genes. (**B**) Clustering of genes and corresponding annotations. Colors represent log_2_(Fold Change) in the snail after exercise.

## Data Availability

The original contributions presented in this study are included in the article/Appendix A. Further inquiries can be directed to the corresponding author(s).

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
