# Peer review of "Rest Induces a Distinct Transcriptional Program in the Nervous System of the Exercised L. stagnalis"

_ijms, 2025, doi:10.3390/ijms26146970_

Round 1

Reviewer 1 Report

Comments and Suggestions for Authors

Major Comments:

Clarify experimental controls and grouping (Lines 545–563):

Detail how control animals were handled in parallel, especially with respect to time and environmental conditions.

Statistical Criteria and Data Interpretation (e.g., Lines 118–170):

The choice to use unadjusted p values (<0.01) due to high variability is understandable but needs stronger justification. Please consider including more stringent FDR thresholds or validating additional transcripts.

Ribosomal ITS Transcripts (Lines 131–154):

The explanation regarding ITS contamination and poly-A-like sequences is plausible but speculative. This section should be shortened or relegated to Supplementary Information unless directly related to your conclusions.

Figures 2–6:

Improve the visual clarity of figures, particularly color coding, axis labels, and gene names.

Clarify figure legends (e.g., Figure 4). What is “ITS” in panels? It may confuse non-specialists.

Functional Inference:

The connection between gene expression shifts and cognitive or behavioral adaptation is highly interesting. However, inferences regarding learning/memory and autism-related genes (e.g., ARID1B, HSPB8, TRRAP) should be more cautious without direct behavioral assays or knockdown validation.

Comparative Cross-Species Analysis (Section 2.4):

This analysis is a strength but currently buried in dense text. Consider summarizing in a new table with fold changes, directions, and significance across species.

Discussion and Hypotheses (Lines 453–535):

The “trajectory model” of gene expression during rest vs. exercise is compelling. Please use a schematic to better visualize your conceptual model.

Minor Comments:

Abstract:

"Such as, homologous genes…" (Line 30) is grammatically incorrect. Rewrite for clarity.

Line 121 (Scatterplot description):

Clearly state axes labels and which genes are included.

Typographical and Formatting Issues:

Several in-line citations have inconsistencies in punctuation and formatting. Ensure consistent citation style per journal requirements.

Line spacing is irregular in many parts of the document.

Comments on the Quality of English Language

The manuscript requires moderate revision for grammar, clarity, and conciseness. Specific issues include:

Grammar and Syntax Issues:

  1. Abstract (Lines 16–33):

    • Use of passive voice is excessive. Recommend simplifying structure, e.g.:

      "We show that two hours of crawling exercise in L. stagnalis alter gene expression. Rest following exercise induces a distinct transcriptional program…"

  2. Introduction:

    • “These changes could play an important role for adaptation…” → should be “in adaptation…”

  3. Awkward Phrasing:

    • Line 70: "had even higher difference" → "showed even greater differences"

  4. Inconsistent Tense:

    • Maintain consistent past tense for results, e.g. "were normalized," not "became normalized."

  5. Article Usage:

    • “The chromatin remodeling and DNA/RNA binding protein…” should be “A chromatin remodeling…”

Recommendations:

  • A thorough professional language edit is advised to improve fluency and style.

  • Avoid jargon where possible and define all acronyms at first mention.

  • Ensure all figure legends are self-contained and grammatically correct.

Author Response

Open Review 1

Comments and Suggestions for Authors

Major Comments:

Clarify experimental controls and grouping (Lines 545–563):

Detail how control animals were handled in parallel, especially with respect to time and environmental conditions.

Response:

Dear Reviewer!

Thank you very much for you time and insightful comments.

We introduced description of the animal handing in the methods section

“Twelve groups of six animals each were used: three groups of active locomotion (exercise, E), three groups of active locomotion followed by rest (exercise-rest, ER) and six control groups (C), which were used for E and ER groups respectively. The snails of E groups were placed in a container with a bottom area of 50x50 cm, filled with a thin (2 mm) layer of water, for 2 hours. The snails of ER groups were treated as above and then returned to aquariums for 2 hours. The control groups were kept in aquariums with a sufficient layer of water that does not change the locomotion regime compared to the usual conditions and were processed at the same time as corresponding experimental animals.  The central ganglia from each group of animals were dissected under 0.1 M MgCl2 anesthesia (0.5 ml per animal for 1 min) immediately after the end of the behavioral procedures and pooled for RNA extraction. “

Statistical Criteria and Data Interpretation (e.g., Lines 118–170):

The choice to use unadjusted p values (<0.01) due to high variability is understandable but needs stronger justification. Please consider including more stringent FDR thresholds or validating additional transcripts.

Response:

We performed a similar annotation analysis limited to the  FDR<0.05 presented in the Supplemental figure 2. In addition, we keep FDR values in the supplemental data. 

Ribosomal ITS Transcripts (Lines 131–154):

The explanation regarding ITS contamination and poly-A-like sequences is plausible but speculative. This section should be shortened or relegated to Supplementary Information unless,⁰ directly related to your conclusions.

Response

Indeed, we do not know the exact reasons for the presence of ITS containing transcripts in our data.  The poly-A primed RT-PCR validation of the transcriptional changes for these transcripts suggests that either polyadenylation is a correct explanation or that these transcripts contain poly-A like sequences.  Irrespective of the exact mechanism, the ITS containing transcripts are mostly repressed by the exercise and become normalised and even slightly induced after rest (Notice how the cloud is turned relative to the protein coding transcripts in Figure 2 and Figure 3). Therefore, it is related to the subject of our paper.

Figures 2–6:

Improve the visual clarity of figures, particularly color coding, axis labels, and gene names.

Clarify figure legends (e.g., Figure 4). What is “ITS” in panels? It may confuse non-specialists.

Response:

We improved Figures 5 and 6 by increasing the font size and limiting the number of presented genes.  We change the ITS in Figure 2 to “ribosomal internal transcribed spacer ”.

We edited Figure 4 legend. We added more detailed description of the Figure 4.

Functional Inference:

The connection between gene expression shifts and cognitive or behavioral adaptation is highly interesting. However, inferences regarding learning/memory and autism-related genes (e.g., ARID1B, HSPB8, TRRAP) should be more cautious without direct behavioral assays or knockdown validation.

Response.

In the discussion we include a statement that our findings require further validation.

“Remarkably, deficiencies of these transcriptional regulators are implicated in the neurodevelopmental defects in mice and humans including TDRD3 [16,17]; SmarcA4 [18,19]; Kmt2a/Mll1 [22], TRRAP [33], AUTS2 transcription factor [75,76] among many others, suggesting that these proteins are pivotal for the neuronal functions and likely impact memory and decision making in L.stagnalis. The functions of these transcripts in L. stagnalis require further experimental validation.”

Comparative Cross-Species Analysis (Section 2.4):

This analysis is a strength but currently buried in dense text. Consider summarizing in a new table with fold changes, directions, and significance across species.

Response:

We added a table that contains top 20 genes repressed or induced by rest or exercise in snails and changed in mice. We moved a supplementary figure 5 to the main figures and made it Figure 7.

Discussion and Hypotheses (Lines 453–535):

The “trajectory model” of gene expression during rest vs. exercise is compelling. Please use a schematic to better visualize your conceptual model.

We made a graphical abstract to illustrate the dynamic changes in gene expression.

Minor Comments:

Abstract:

"Such as, homologous genes…" (Line 30) is grammatically incorrect. Rewrite for clarity.

Response:

“Our findings are likely relevant to the physiology of exercise, rest and learning in other species. For example, chronic voluntary exercise training in mice affects expression of many homologous genes in the hippocampus. “

Line 121 (Scatterplot description):

Clearly state axes labels and which genes are included.

Response:

“In colors are genes regulated by exercise and exercise-rest ( FC>2 or FC<0.5, p<0.01) and in grey are genes whose changes are not significant. “

Typographical and Formatting Issues:

Several in-line citations have inconsistencies in punctuation and formatting. Ensure consistent citation style per journal requirements.

Response.

We double checked the citation.

Line spacing is irregular in many parts of the document.

Comments on the Quality of English Language

The manuscript requires moderate revision for grammar, clarity, and conciseness. Specific issues include:

Grammar and Syntax Issues:

Abstract (Lines 16–33):

Use of passive voice is excessive. Recommend simplifying structure, e.g.:

"We show that two hours of crawling exercise in L. stagnalis alter gene expression. Rest following exercise induces a distinct transcriptional program…"

Response:

Thank you. We changed the abstract a little bit.:

“Abstract: In the freshwater snail L. stagnalis, two hours of shallow water crawling exercise are accompanied by the formation of memory, metabolic, neuronal and behavioral changes such as faster orientation in a novel environment. Interestingly, rest following exercise enhances  serotonin and dopamine metabolism linked to formation of memory and adaptation to novel conditions. However, the underlying transcriptional responses are not characterized. Here we show that, while two hours of L. stagnalis forced crawling exercise produce significant changes in nervous system gene expression, the subsequent rest induces a completely distinct transcriptional program. “

“These changes could play an important role for adaptation…” → should be “in adaptation…”

Awkward Phrasing:

Line 70: "had even higher difference" → "showed even greater differences"

Inconsistent Tense:

Maintain consistent past tense for results, e.g. "were normalized," not "became normalized."

Article Usage:

“The chromatin remodeling and DNA/RNA binding protein…” should be “A chromatin remodeling…”

Response.

All my life I have been trying to better understand articles over and over again…

Recommendations:

A thorough professional language edit is advised to improve fluency and style.

Avoid jargon where possible and define all acronyms at first mention.

Ensure all figure legends are self-contained and grammatically correct.

Submission Date

06 June 2025

Date of this review

21 Jun 2025 21:52:47

Reviewer 2 Report

Comments and Suggestions for Authors

Authors have analyzed transcriptional changes in snail L. stagnalis at rest and after exercised followed by rest condition. I found this is a unique way to understand how neurological transcriptome changes after exercise and then resting, especially by using snails. The authors have used appropriate methodology and demonstrated it well. All the sections are well written and presented appropriately. However, I have some minor comments which need to be addressed:

  1. Figure 3: The graph is a very old fashioned excel format. Please redraw this with a clear background.
  2. Figure 3: There are couple of control groups in which only two animals have been represented (black dots). If there are only two animals, it is incorrect to apply any statistics on these datasets.
  3. Define EF1a and ITS. Please make sure the abbreviations are properly defined.
  4. Line 135-136, Line 472-473: Missing period at the end of a sentence
  5. I would recommend a thorough grammar check for any typos.

Author Response

Open Review 2

Comments and Suggestions for Authors

Authors have analyzed transcriptional changes in snail L. stagnalis at rest and after exercised followed by rest condition. I found this is a unique way to understand how neurological transcriptome changes after exercise and then resting, especially by using snails. The authors have used appropriate methodology and demonstrated it well. All the sections are well written and presented appropriately. However, I have some minor comments which need to be addressed:

Figure 3: The graph is a very old fashioned excel format. Please redraw this with a clear background.

Response:

Dear Reviewer!

Thank you very much for you time and insightful comments.

Thank you very much for your comment. We redraw this figure.

Figure 3: There are couple of control groups in which only two animals have been represented (black dots). If there are only two animals, it is incorrect to apply any statistics on these datasets.

Response:

We included the number of animals in the graph. These are just overlapping spots, notice how averaged values are skewed from the center for these data.

Define EF1a and ITS. Please make sure the abbreviations are properly defined.

Response:

We corrected this in the methods section.

“..and dCt  method was used to estimate the quantities of tested genes relative to the reference gene elongation factor 1-alpha (EF1a) [146].”

Line 135-136, Line 472-473: Missing period at the end of a sentence

I would recommend a thorough grammar check for any typos.

Response:

Thank you very much.

Submission Date

06 June 2025

Reviewer 3 Report

Comments and Suggestions for Authors

Exercise activates brain function in vertebrates and invertebrates alike, improving memory and promoting neurogenesis. These benefits may be mediated by neuro-regulatory and neurotrophic systems. This study investigates transcriptional changes in the nervous system of L. stagnalis after exercise and rest to reveal its molecular mechanisms and cross-species relevance. While this study provides new insights into the relationship between exercise, rest, and neurodevelopmental disorders, such as autism, several important questions remain unanswered. These issues are listed below:

  1. Some experimental data (e.g., RT-PCR validation) do not report specific sample sizes.
  2. Some figures have low resolution, which affects readability.
  3. The discussion concludes that "rest activates neurodevelopmental genes," but lacks functional validation data (e.g., gene knockdown experiments). To avoid overinterpretation, it is recommended that the conclusion include the statement, "These findings require further validation through functional experiments."
  4. The cross-species conservation analysis is based solely on homology comparisons and has not experimentally validated changes in the expression of homologous genes in mice.
  5. The biological significance of "differential expression of ribosomal genes" is not adequately explained; only speculation regarding its association with energy expenditure is provided. Supplementing with literature support is recommended, such as citing studies on ribosomal RNA regulation and stress responses in other species.
  6. Some sentences contain grammatical errors. It is recommended that a full review of the entire text be conducted to correct the grammar.

Author Response

Open Review 3

Comments and Suggestions for Authors

Exercise activates brain function in vertebrates and invertebrates alike, improving memory and promoting neurogenesis. These benefits may be mediated by neuro-regulatory and neurotrophic systems. This study investigates transcriptional changes in the nervous system of L. stagnalis after exercise and rest to reveal its molecular mechanisms and cross-species relevance. While this study provides new insights into the relationship between exercise, rest, and neurodevelopmental disorders, such as autism, several important questions remain unanswered. These issues are listed below:

Some experimental data (e.g., RT-PCR validation) do not report specific sample sizes.

Some figures have low resolution, which affects readability.

Response.

Dear Reviewer!

Thank you very much for you time and insightful comments.

We included the observation numbers in the figure,

We improved the readability of Figures 5 and 6 by increasing font size and limiting the number of genes.

The discussion concludes that "rest activates neurodevelopmental genes," but lacks functional validation data (e.g., gene knockdown experiments). To avoid overinterpretation, it is recommended that the conclusion include the statement, "These findings require further validation through functional experiments."

 Response:

We included this phrase in the discussion.

“Remarkably, deficiencies of these transcriptional regulators are implicated in the neurodevelopmental defects in mice and humans including TDRD3 [16,17]; SmarcA4 [18,19]; Kmt2a/Mll1 [22], TRRAP [33], AUTS2 transcription factor [75,76] among many others, suggesting that these proteins are pivotal for the neuronal functions and likely impact memory and decision making in L.stagnalis. The functions of these transcripts in L.stagnalis require further experimental validation.”

The cross-species conservation analysis is based solely on homology comparisons and has not experimentally validated changes in the expression of homologous genes in mice.

Response:

Agree. The data of Methi 2024, which we used for our analysis, is based on the single cell sequencing of mice neurons after exercise and does not include PCR or functional validation.

The biological significance of "differential expression of ribosomal genes" is not adequately explained; only speculation regarding its association with energy expenditure is provided.

Supplementing with literature support is recommended, such as citing studies on ribosomal RNA regulation and stress responses in other species.

Response.

We added text to the results, section:

2.3.2 Annotation of genes induced or repressed upon rest.

Rest represses expression of several translation related proteins including DDX56, which is involved in expression of ribosomal RNAs in stem cells [111], Peptidyl-tRNA hydrolase MRPL58 also known as ICT1, a mitochondrial ribosomal protein, functioning in translation termination and also involved in the suppression of apoptosis [112]. Rest also downregulates Nip7, which is required for the 18S rRNA maturation [113]. In addition, it was recently reported that depletion of above-mentioned rest repressed DYRK1A, causing global reduction of the ribosomal proteins [114]. Altogether, this suggests that protein translation is likely repressed after rest.

And to the Discussion:

 “Unexpectedly, a distinct subset of genes with homology to the ribosomal internal transcribed spacer was repressed after exercise and normalised after rest. In turn,  a distinct set of transcripts encoding proteins involved in translational regulation were repressed following rest. The functional significance of this response is yet to be understood. It is known that an eukaryotic cell is able to quickly respond to a sudden stress by the rapid suppression of protein synthesis [138,139]. Ribosomal protein biosynthesis is regulated by the nutrient availability in distant organisms. For instance, the conserved rapamycin (TOR) signal-transduction pathway adjusts ribosomal protein biosynthetic capacity to nutrient availability in yeasts and mammals [138].

In neurons, a number of mechanisms modulate ribosomal functions [140].  For example, changes of the local ribosomal activity in distal cellular locations allows remodeling of the local proteome, underlying dendritic changes and neuronal plasticity [140]. 

During the periods of high energy release like exercise, the neuron may try to stop additional energy-consuming processes including protein synthesis. The rebound effect observed after the exercise agrees with this suggestion. Interestingly, more than 50 years ago, an inhibition of mRNA synthesis followed by rebound was seen in response to the electric stimulation of neurons in L. stagnalis [141]. The genes encoding ribosomal proteins were downregulated by incubation of hippocampal neuronal culture in picrotoxin, which causes a strong excitation [142]. The authors suggested that this may indicate redistribution of transcriptional resources. Another possibility is that during the periods of high risk for DNA stability, the decrease in the DNA transcription and RNA translation may help to protect the neuronal DNA [143]. The same may explain downregulation of translation related genes upon rest. “

Some sentences contain grammatical errors. It is recommended that a full review of the entire text be conducted to correct the grammar.

Response.

We re-examined the text for grammatical errors and punctuation.

Submission Date

06 June 2025

Date of this review

23 Jun 2025 02:12:54